# In Vitro Bioaccessibility and Antioxidant Activity of Polyphenolic Compounds from Spent Coffee Grounds-Enriched Cookies

**DOI:** 10.3390/foods10081837

**Published:** 2021-08-09

**Authors:** Luigi Castaldo, Sonia Lombardi, Anna Gaspari, Mario Rubino, Luana Izzo, Alfonso Narváez, Alberto Ritieni, Michela Grosso

**Affiliations:** 1Department of Pharmacy, University of Naples “Federico II”, 49 Domenico Montesano Street, 80131 Naples, Italy; luigi.castaldo2@unina.it (L.C.); sonia.lombardi@unina.it (S.L.); annagaspari@virgilio.it (A.G.); alfonso.narvaez@unina.it (A.N.); alberto.ritieni@unina.it (A.R.); 2InterKom S.p.A., 20 Gian Lorenzo Bernini Street, 80129 Naples, Italy; rubino.mario@hotmail.it; 3Department of Molecular Medicine and Medical Biotechnology, School of Medicine, University of Naples “Federico II”, CEINGE-Biotecnologie Avanzate, 80131 Naples, Italy; michela.grosso@unina.it

**Keywords:** valorization, antioxidant activity, food applications, food waste, bioactive molecules, polyphenols, bioaccessibility

## Abstract

Spent coffee ground (SCG) is a significant by-product generated by the coffee industry. It is considered a great source of bioactive molecules well-recognized for exerting biological properties. This study aimed to implement SCG in a baked foods, such as cookies (SCGc), to increase their bioactive potential. A comprehensive study of the polyphenolic fraction of the SCG and SCGc using a high-resolution mass spectrometry analysis was performed. Moreover, the polyphenol bioaccessibility and change in antioxidant activity during simulated gastrointestinal digestion (GiD) were assessed. Data showed that SCGc provided 780 mg of melanoidins, 16.2 mg of chlorogenic acid (CGA), 6.5 mg of caffeine, and 0.08 mg of phenolic acids per 100 g of sample. Moreover, the 5-caffeoylquinic acid was the most relevant CGA found in SCG (116.4 mg/100 g) and SCGc (8.2 mg/100 g) samples. The antioxidant activity evaluated through three spectrophotometric tests, and the total phenolic compounds of SCGc samples exhibited significantly higher values than the control samples. Furthermore, during simulated GiD, the highest bioaccessibility of SCGc polyphenols was observed after the colonic stage, suggesting their potential advantages for human health. Therefore, SCG with high content in bioactive molecules could represent an innovative ingredient intended to fortify baked food formulations.

## 1. Introduction

Global coffee production reached 171.9 million of 60 kg bags (International Coffee Organization) in the year 2020/2021 [1]. As a result, large quantities and varieties of waste and by-products are inevitably generated by the coffee industry, among which the spent coffee ground (SCG) represents the most significant product (45%), with a yearly production of about 6 million tons [2,3]. To date, suggested use for SCG includes the production of biofuel, biosorbents, animal feed, and composts [4]. Notwithstanding that, due to the high level of bioactive molecules contained in SCG, many scientific studies have focused on finding a way to exploit this coffee waste in the nutraceutical and food industry as a natural source of antioxidants [5]. Indeed, it has been reported that SCG contains a broad range of active compounds, which include dietary fiber, high molecular weight melanoidins (HMWM), alkaloids, and phenolic compounds [6]. These active molecules are well-recognized for exerting various biological properties involved in the reduction of age-related diseases with potential interest in the formulation of functional foods, fortified foods, and dietary supplements [7,8].

Polyphenols are phytochemicals found in many plant-based foods, such as wine, tea, vegetables, fruits, cocoa, and coffee, among others [9]. Several studies reported that these secondary metabolites may exert diverse bioactivities, such as antioxidant, antiviral/antibacterial, anticarcinogenic, and anti-inflammatory activities [10]. In this line, polyphenol-rich diets seem to be protective against several diseases, including type 2 diabetes, Parkinson’s disease, obesity, cardiovascular disease, and cancer [11]. Among polyphenolic compounds, several investigations reported that chlorogenic acids (CGAs) represent the main dietary polyphenols present in SCG at a concentration level similar to that found in coffee drink samples [12]. In their structure, all CGAs include a common residue of quinic acid esterified with a hydroxycinnamic acid, namely *p*-coumaric acid, ferulic acid, and caffeic acid, giving rise to *p*-coumaroylquinic acid (*p*CoQA), feruloylquinic acid (FQA), caffeoylquinic acid (CQA), and dicaffeoylquinic acid (diCQA) [13]. The isomers of CQA are the most abundant polyphenols reported in both SCG and coffee brew samples [14,15]. These important molecules are well-known to display potent antioxidant power capable of promoting human health benefits [16,17].

To exert therapeutic activity, polyphenols need to be bioaccessible to some extent for absorption into target tissues [18]. Several in vitro digestion models simulating fermentation of human gut microbiota have been developed to assess polyphenol bioaccessibility and antioxidant activity [19]. It has been reported that the combined action of Pronase (mixture of bacterial protease) and Viscozyme L (commercial mix of several carbohydrases) is able to reproduce the intestinal fermentation [20,21]. Gut microbiota seems to play a fundamental role in polyphenols adsorption [22]. Previous studies have reported that around two-thirds of the CGA in coffee reach the small and the large intestine intact, where they could be metabolized by the intestinal microbiota, achieving a positive effect in maintaining human health [23,24,25]. Moreover, SCG contains melanoidins which, after ingestion, act as dietary fibers, escaping digestion to be fermented by the colonic microbiota [26]. As reported by many investigations, melanoidin can exert high antioxidant activity due to the presence of CGAs residue incorporated into them during the coffee-roasting process, which can be released by colon fermentation [27].

Habitual dietary intake of fibers is recognized to play a protective role against obesity, colon cancer, and diabetes [28]. SCG with approximately 51% of crude dietary fibers, previously classified as antioxidant dietary fiber, may serve as an innovative source to develop fortified foods rich in dietary fibers [29]. Furthermore, caffeine represents the major alkaloid recovered from SCG samples [30]. It is considered one of the most active molecules in coffee, having a positive effect on the central nervous system, helping to improve cognitive and physical performance [31].

In recent years, many baked food products have been enriched with functional ingredients from selected food by-products, including tomato by-products, pomegranate peel, apple, and sour cherry pomaces, in order to enhance their functional properties or as natural additives [32,33,34]. Although the use of SCG to increase the biological properties of baked food has been reported, the metabolic fate of the polyphenols released by SCG-based foods during an in vitro gastrointestinal digestion (GiD) has been barely investigated [35,36]. Martinez-Saez et al. [35] evaluated the use of SCG material as a food ingredient to fortify bakery products, investigating the microbiological safety and sensory attributes in cookies. These developed cookies showed an acceptable microbiological profile and improved sensory attributes compared with the commercially available cookies. The authors concluded that SCG could be considered a suitable food ingredient for fortified bakery production. Another scientific work [36] investigated the use of the antioxidant compounds extracted from SCG as an ingredient in innovative bakery products. The developed formulation showed increased antioxidant activity, total dietary fiber, and phenolic bioaccessibility after in vitro GiD excluding the colonic phase in the analysis. However, some scientific evidence reported that the large intestine may represent the most important physiological site of action for the coffee-related antioxidant compounds [37]. 

Therefore, the aim of this study was to implement SCG in a baked food, such as cookies, to increase its antioxidant activity and polyphenolic compound content to create innovative, healthy food. Moreover, a complete simulated GiD was used to assess the SCGc polyphenol bioaccessibility as well as the variations in the antioxidant activity in order to evaluate the application of SCG in the development of healthy bakery products.

## 2. Materials and Methods

### 2.1. Reagents and Materials

Polyphenol standards (purity > 98%) were purchased as follows: 4-CQA, 3,4-diCQA, *p*-coumaric acid, quinic acid, ferulic acid, caffeic acid, and gallic acid from Sigma-Aldrich (Milan, Italy). The reagents used in simulated GiD, including α-amylase from human saliva, protease from Streptomyces griseus (Pronase), pepsin from porcine gastric mucosa, bile salts, Viscozyme L, pancreatin from porcine pancreas, potassium chloride (KCl), sodium chloride (NaCl), potassium dihydrogen phosphate (KH_2_PO_4_), sodium bicarbonate (NaHCO_3_), magnesium chloride hexahydrate (MgCl_2_(H_2_O)_6_), ammonium carbonate (NaHCO_3_)_2_), sodium hydroxide (NaOH), and calcium chloride dihydrate (CaCl_2_(H_2_O)_2_), were provided from Sigma-Aldrich (Milan, Italy). The standards used in antioxidant assays, including 2,3,5-triphenyltetrazolio chloride (TPTZ), Trolox, 2’2-azino-bis-3-ethylbenzthiazoline-6-sulphonic acid (ABTS), 1,1-diphenyl-2-picrylhydrazyl (DPPH), potassium persulfate (K_2_S_2_O_8_), and ferric chloride (FeCl_3_), were acquired from Sigma-Aldrich (Milan, Italy).

Formic acid (FA), water (H_2_O), and methanol (MeOH) were provided from Carlo Erba reagents (Milan, Italy). Deionized H_2_O was obtained from a Milli-Q water system (Millipore, Darmstadt, Germany).

### 2.2. Sampling and Coffee Preparation

Colombian roasted coffee (*C. arabica* L.; medium degree) beans (*n* = 10) were acquired from a local supermarket in Italy. The roasted coffee samples were finely ground by using a coffee grinder (Bosch Elettrodomestici, TSM6A013B, Milan, Italy) to obtain a composite sample. Coffee was prepared through an American coffeemaker (Aigostar Chocolate 30HIK, Aigostar S.r.L., Milan, Italy), and then, SCG was recovered from the filter and dried in a laboratory oven until the moisture of the material reached a level between 12% and 14.5%. Finally, the samples were stored at room temperature.

### 2.3. Spent Coffee Grounds-Enriched Cookies Preparation

Spent coffee grounds-enriched cookies (SCGc) were prepared following the recipe reported previously by Colantuono et al. [38]. Briefly, dough was prepared with 5 g of SCG, 40 g of wheat flour, 10 g of butter, 17.5 g of sugar, 0.2 g of sodium chloride, 0.8 g of bicarbonate, and 20 g of H_2_O. Control cookies (CTc) were prepared using wheat flour in substitution of the SCG. The doughs were molded into circular shapes of a diameter of 4 cm and 5 mm in height. After that, the cookies (samples and controls) were baked simultaneously at 205 °C for 10 min in an oven. Then, the samples were cooled and stored at room temperature in bags of polyethylene until the analysis.

### 2.4. Polyphenols and Caffeine Extraction

Polyphenols and caffeine extraction was carried out according to the procedure previously reported by Gonçalves et al. [14]. Briefly, 300 mg of samples were suspended in 25 mL of H_2_O:EtOH 75:25 (*v/v*), stirred for 15 min at 300× *g*, and then sonicated for 15 min. After that, the mixture was centrifuged at 4200× *g* for 5 min, and the supernatant was collected and employed for the total phenolic content (TPC), antioxidant activity determination, and spectrometric characterization.

### 2.5. High Molecular Weight Melanoidins Content

High molecular weight melanoidins quantification was assessed following the procedure previously reported by De Cosío-Barrón et al. [39]. In short, Amicon Ultra-4-cell model regenerated cellulose with 10 kDa of nominal molecular mass were used to ultrafiltrate 4 mL of extract (5 mg/mL). The samples were subjected to ultrafiltration for 80 min at 5000× *g*. The retentates were washed three times with 4 mL of water. HMWM was quantified by weighing the freeze-dried retentate resulted after dialysis. The obtained results were displayed as g/100 g.

### 2.6. Ultra-High-Performance Liquid Chromatography and Orbitrap High-Resolution Mass Spectrometry Analysis

The separation of the investigated analytes was obtained using an ultra-high-performance liquid chromatography (UHPLC; Dionex Ultimate 3000, Thermo Fischer Scientific, Waltham, MA, USA), provided by an autosampler device, a degassing system, a Quaternary UHPLC pump, and a Kinetex column F5 (50 mm × 2.1 mm, 1.7 µm particle size, Phenomenex, Torrance, USA) thermostated at 25 °C. The mobile phases (phase A: water; and phase B: methanol) were both prepared at 0.1% of formic acid. The separation gradient program started as follows: initial 0% B for 1 min and then rose up to 80% B in 2 min. Afterward, the gradient increased again to 100% B in 3 min. Then, the gradient returned to the equal % B in 2 min and was maintained for 2 min for column re-equilibration.

A Q-Exactive mass spectrometer (Thermo Fischer Scientific, Waltham, MA, USA) combined with an electrospray (ESI) source allowed the acquisition in negative/positive ion mode fast polarity-switching mode, setting two scan events full ion MS and all ion fragmentation (AIF). The following parameters were set in full MS experiments: maximum injection time, 200 ms; automatic gain control (AGC) target, 1 × 10^6^; scan range, 80–1200 *m/z*; microscans, 1; mass resolution, 35,000 full width at half maximum (FWHM); scan time, 0.10 s; retention time to 30 s; and isolation window to 5 *m/z*. The collision energies (CEs) were optimized considering values varied in the range 10–60 eV. Identification was based on exact mass measurements with a mass error < 5 ppm in both full ion MS and AIF mode. Data processing was carried out through Quan/Qual Browser Xcalibur software 3.1.66.19 (Thermo Fischer Scientific, Waltham, USA).

### 2.7. In Vitro Gastrointestinal Digestion

The SCGc and CTc samples were subjected to the in vitro GiD process following the developed protocol recently created by the INFOGEST network. The simulated salivary (SSF), gastric (SGF), and intestinal (SIF) fluids were developed in accordance with procedure reported by Minekus et al. [40] (Appendix A).

In order to simulate the oral condition, 500 mg of the grinded samples were mixed with 500 µL of the α-amylase solution, 3.5 mL of SSF, 25 µL of 0.3 M of calcium chloride solution, and 0.975 mL of H_2_O. Then, the pH of the solution was adjusted to 7 with sodium hydroxide 1 M and incubated at 37 °C for 2 min.

The gastric phase was simulated by adding to the mixture 0.685 mL of H_2_O, 5 µL of 0.3 M of calcium chloride solution, and 1.6 mL of pepsin solution. Afterward, the pH of the solution was adjusted to 3 with HCL 1 M and incubated at 37 °C for 2 h.

Then, 1.3 mL of H_2_O, 5 mL of pancreatin solution, 2.5 mL of bile salt solution, and 40 µL 0.3 M calcium chloride solution were added to the mixture in order to recreate the intestinal condition. After, the pH of the solution was adjusted to 7 with sodium hydroxide 1 M and incubated at 37 °C for 120 min.

Finally, to simulate the activity of gut microbiota, the samples were subjected to the previously described protocol [20]. In brief, the pH of the solution was adjusted to 8 with sodium hydroxide 1 M, and then, 5 mL of Pronase solution at a concentration level of 5 mg/mL was added. The samples were incubated for 1 h at 37 °C. Then, 150 µL of Viscozyme L and 5 mL of water were added to the mixture. The pH of the solution was adjusted to 4 with HCl 1 M and incubated to 16 h at 37 °C.

An aliquot of the supernatant (1 mL) was recovered at the end of each phase of in vitro GiD and replaced by the appropriate fluid phase in order to assess the changes in polyphenol bioaccessibility and antioxidant activity during the different stages of the GiD.

### 2.8. Determination of the Antioxidant Activity

The antioxidant activity of the cookies (digested and not-digested) and SCG material was evaluated by using three different assays, including DPPH, FRAP, and ABTS tests.

#### 2.8.1. DPPH Assay

The determination of DPPH assay was carried out following the procedure reported by Dini et al. [41]. In short, 1 mg of DPPH standard was dissolved in methanol until reaching a value of absorbance of 0.90 (±0.01) at 517 nm. Afterward, 200 µL of sample extract were added to 1 mL of DPPH solution. The absorbance value after 10 min was immediately recorded.

#### 2.8.2. FRAP Assay

The FRAP method was conducted according to the procedure described by Izzo et al. [42]. As reported, the FRAP solution was prepared by adding 2.5 mL of acetate buffer, 0.25 mL of TPTZ in HCL, and 0.25 mL of a 20 mM of FeCl_3_ solution. Afterward, 150 µL of sample extract were added to 2.85 mL of FRAP solution. The absorbance value at 593 nm after 4 min was immediately recorded.

#### 2.8.3. ABTS Assay

The determination of ABTS activity was assessed by using the procedure reported by Dini et al. [43]. In short, 5 mL of ABTS (7 mM) were mixed to 88 µL of K_2_S2O_8_ (2.5 mM) and kept at room temperature for 16 h. Afterward, the EtOH was used to dilute the ABTS solution until reaching a value of absorbance of 0.70 (±0.01) at 734 nm. Then, 100 µL of sample extract were added to 1000 µL of ABTS solution. The absorbance value after 3 min was immediately recorded.

### 2.9. Determination of Total Phenolic Content

The Folin–Ciocalteu procedure was performed to assess the TPC content in accordance with the methodology described previously by Izzo et al. [44]. Briefly, 125 µL of the Folin–Ciocalteu reagent (2 N) were added to 500 µL of H_2_O and mixed with 125 µL of sample extract. The mixture was incubated for 6 min at room temperature. Then, 1.25 mL of NaCO_3_ solution (7.5%) and 1 mL of H_2_O were added. The absorbance value after 90 min at 760 nm was immediately recorded. 

### 2.10. Statistics and Data Analysis

Tukey’s test was used to evaluate differences between SCGc and control samples considering *p*-value less than 0.05 as significant. All analysis were conducted in triplicate and the results expressed as average ± standard deviation (SD). Data processing was carried out through Stata 12 software (StataCorp LP, College Station, TX, USA).

## 3. Results

### 3.1. High Molecular Weight Melanoidins Content

The HMWM content in SCG and SCGc samples was carried out through the ultrafiltration technique. As far as SCG was concerned, the HMWM were quantified at a concentration range from 10.15 to 11.47 g/100 g, with an average value of 10.80 g/100 g. Moreover, HMWM levels found in SCGc samples ranged from 0.75 to 0.84 g/100 g, with an average value of 0.78 g/100 g.

### 3.2. Identification of Polyphenol Compounds and Caffeine in the Assayed Samples Using UHPLC-Q-Exactive Orbitrap

Identification of individual hydroxycinnamic acids (*n =* 4), CGAs (*n* = 9) and caffeine in the SCG, SCGc, and CTc samples was carried out through UHPLC-Q-Orbitrap high-resolution mass spectrometry. Good separation of the assayed analytes was achieved in 13 min. Nevertheless, the isomers 4-FQA and 5-FQA were quantified together (4+5-FQA), caused by insufficient chromatographic separation. The chemical formula, ion assignment, retention time (RT), and measured and theoretical mass for the studied analytes are reported in Table 1. The structural isomers *p*-CoQA (*m/z* 337.09289), CQA (*m/z* 353.08780), and diCQA (*m/z* 515.11950) were identified by comparing the obtained fragmentation pattern with data previously reported [45] and the RT of standards with the obtained peaks.

### 3.3. Quantification of Polyphenol Compounds and Caffeine in the Assayed Samples Using UHPLC-Q-Exactive Orbitrap

The predominant CGAs, some important phenolic acids, and caffeine were quantified in SCG, SCGc, and CTc samples by using a high-resolution Orbitrap mass analysis. Calibration curves with ten concentration levels (regression coefficient > 0.99) were carried out for the quantitative determination of the found molecules.

Up to thirteen different polyphenolic compounds were quantified in the assayed samples, as reported in Table 2. Total CQA represented from 84.8 to 85.8% of total CGA detected in SCG and SCGc samples, respectively. Moreover, 5-CQA showed to be the most relevant CGA, being quantified in SCG and SCGc samples at a mean value of 1163.9 and 81.6 mg/kg, respectively. Referring to FQAs isomers, 3- and 4+5-FQA represented from 7.9 (SCG) to 8.4% (SCGc) of total CGA found in the here-investigated samples. As far as diCQAs concerned, 3,5-diCQA was the compound quantified at the highest concentration in both SCG and SCGc samples, at an average value of 135.9 and 8.3 mg/kg, respectively. Furthermore, *p*CoQA isomers mainly represented by 3- and 5- *p*CoQA were detected as the minor CGAs found in both SCG and SCGc samples at a mean value of 4.75 and 0.26 mg/kg, respectively. Moreover, some bioactive phenolic acids (ferulic acid, caffeic acid, quinic acid, and *p-*coumaric acid) were assessed in the analyzed samples. These important molecules represented from 0.42 to 0.46% of total polyphenolic compounds found in samples. Caffeic acid was the most common one, ranging from 0.5 (SCGc) up to 7.2 (SCG) mg/kg. Apart from polyphenols, caffeine was also evaluated in the here-assayed samples. As shown in Table 2, SCG samples displayed a caffeine concentration up to 1193.89 mg/kg, whereas the content detected in SCGc samples was 64.60 mg/kg.

### 3.4. In Vitro Bioaccessibility of Coffee Polyphenols

Simulated GiD was performed in order to evaluate the SCGc polyphenols bioaccessibility and variation in antioxidant activity. In each step of the GiD, the content in TPC was assessed by using the Folin–Ciocalteu method.

As highlighted by Table 3, SCGc samples showed significantly (*p*-value < 0.05) higher TPC values than CTc samples in each step of the GiD. Moreover, along the simulated GiD, the colonic stage (Pronase plus Viscozyme L stages) showed the highest TPC value, 167.7 and 116.8 mg GAE/100 g for SCGc and CTc samples, respectively. In particular, the potential polyphenol bioaccessibility in the colonic stage reported by SCGc samples was 96.7%, whereas for Ctc samples, it was 88.7%.

The antioxidant activity of the not-digested and digested samples was assessed using three different tests: DPPH, FRAP, and ABTS assays. Table 4 displays the results as mmol of Trolox equivalent (TE) per kilogram of the sample (average value and SD).

Concerning the antioxidant activity measured in not-digested samples, SCGc showed higher antioxidant activity than CTc samples in all evaluated spectrophotometric tests. In particular, not-digested SCG samples showed a percentage of increase in antioxidant activity of 12.4%, 12.9%, and 24.7% for DPPH, FRAP, and ABTS respectively, when compared to not-digested CTc.

On the other hand, the variation in antioxidant activity release during the simulated GiD of the SCGc samples was also evaluated. Digested samples (SCGc and CTc) showed significantly lower antioxidant activity (*p*-value < 0.05) than the not-digested samples through all the simulated GiD stages. However, compared to CTc samples, data highlighted that the SCGc samples showed a higher antioxidant activity in all simulated GiD phases. Moreover, the colonic stage (considered as Pronase plus Viscozyme L stages) showed the highest antioxidant activity along the simulated GiD.

## 4. Discussion

In this study, a comprehensive characterization of polyphenols compounds and caffeine contained in SCG and SCG-enriched cookies was carried out using an UHPLC-Q-Exactive Orbitrap instrument. In detail, nine predominant CGAs, four phenolic acids, and caffeine were assessed in SCG as well as SCGc samples.

Overall, the results indicate that SCG material may represent an important source of bioactive compounds, such as high content of polyphenols, melanoidins, and caffeine. Regarding the CGAs content found in assayed samples, the total concentration displayed by SCG samples was 2465.6 mg/kg. These levels showed a two-fold increase compared to SCG samples previously analyzed by Angeloni et al. [12], who reported total CGAs concentration up to 1299.8 mg/kg. The monitored CGAs were only three (5-CQA, 3-, and 5-diCQA) of nine CGAs studied in the analyzed SCG samples, which may explain the different levels observed. In addition, these variabilities in the concentration of CGAs could be attributed to the influence of many factors, including origin of SCG, brewing procedures, and roasting degree, which plays a fundamental role in the presence of CGAs in this studied coffee by-product [46]. Furthermore, the data obtained clearly showed that 5-CQA was the most prevalent CGA in the analyzed SCG samples, accounting for 84% of total CGAs. A wide variability was observed by Campos et al. [5], who reported a concentration range of 5-CQA from 397 to 2642 mg/kg in SCG samples. In the last decade, a broad number of epidemiological and experimental studies have linked CGA habitual intake to specific biological effects involved in preventing degenerative diseases and maintaining human health status [47,48]. This is due to the several properties that have been reported for these active molecules, playing a fundamental role in modulating lipid and glucose metabolism, helping to handle a wide range of disorders such as diabetes, hepatic steatosis, obesity, and cardiovascular disease as well [49,50,51].

As regards the melanoidins content in the assayed SCG samples, the levels found (~11 g/100 g) in the investigated samples were lower when compared to SCG samples previously analyzed, reporting a concentration range between 13 to 25 g/100 g [52]. Melanoidins are well recognized as important heterogeneous compounds found in SCG material able to exert biological activities [53]. According to Moreira et al. [54], melanoidins have strong antioxidant activity, mainly due to the presence of CGA residue incorporated into them during the coffee-roasting process.

Concerning the caffeine occurrence in SCG material, our results revealed high concentrations in this mentioned alkaloid reaching up to 1193.89 mg/kg. Similar high levels were reported by Cruz et al. [55], in which caffeine was found in espresso SCG material at concentrations ranging from 800 to 1400 mg/kg. Available evidence showed that moderate caffeine consumption (<400 mg per day) appears to be related to potentially positive effects in healthy adults, helping to reduce the incidence of various chronic diseases and improving mental and physical performances [56].

On the other hand, SCGc prepared in this study at 7.5% of SCG provided 780 mg of melanoidins, 16.2 mg of CGAs, 6.5 mg of caffeine, and 0.08 mg of phenolic acids per 100 g of SCG-enriched cookies. This formulation guaranteed microbiological and chemical safety (hydroxymethylfurfural and acrylamide) of the product according to Martinez-Saez et al. [35], who evaluated the application of SCG in the formulation of SCG-enriched cookies. Moreover, the authors reported that the taste, texture, colour, and overall acceptance of SCG-enriched cookies were comparable to commercial cookies. The main aim of this study was to evaluate the bioaccessibility of polyphenols from SCG-enriched baked food as well as their antioxidant activity displayed during an in vitro GiD to evaluate the application of SCG material as an innovative ingredient in the development of healthy bakery products. In the last decades, SCG material has been successfully employed to produce new foods and beverages, including spirits, pastry, cereal, and confectionery [3]. As reported, the products developed with SCG were appropriate for particular nutritional needs due to the high dietary fiber content and low glycemic and caloric index.

Overall, our results showed that the TPC value and antioxidant activity significantly increased (*p* < 0.05) in biscuits formulated with SCG. Different findings have been previously reported in baked biscuits enriched with dietary fiber extracted from SCG [36]. The monitored TPC value in the enriched cookies was similar to the control. The authors revealed that dietary fiber extracted from SCG retained only 50% of phenolic compounds present in SCG material, resulting in a non-significant contribution of dietary fiber extracted from SCG to the antioxidant properties and TPC level of the formulated cookies.

The in vitro GiD was performed following the protocol recently developed by the INFOGEST network until the duodenal phase, whereas Viscozyme L and Pronase were used to simulate the activity of gut microbiota. Pronase is a commercial mixture of several bacteria nonspecific exo- and endoproteases, while Viscozyme L is a multi-enzyme complex containing a broad range of carbohydrases, such as arabanase, β-glucanase, xylanase, and cellulase. Previous articles have reported that the combined use of these commercial products in in vitro studies represents an effective alternative to the conventional use of the fecal inoculum to reproduce gut microbial metabolism [20,21].

The here-obtained results showed that both the TPC values and the antioxidant activity of SCGc samples exhibited significantly higher values than CTc samples during each step of the simulated GiD. Compared to CTc samples, the highest percentage of increase in antioxidant activity and TPC value was observed during the colonic phase in SCGc samples. In particular, the TPC value observed in SCGc samples was increased by about 30% at the end of the colonic phase, whereas SCGc samples displayed a two-fold increase in antioxidant activity after the colonic phase than the CTc samples. These outcomes suggested that the gut microbiota might be able to release phenolic compounds from SCGc samples with enhanced antioxidant activity, resulting in positive implications in improving and maintaining human health.

The increased value of TPC and antioxidant activity in SCGc samples after the colonic stage may be due to the presence of melanoidins in SCGc formulation. Many scientific studies have suggested that melanoidins have an important role in the gastrointestinal tract [57]. It has been reported that the melanoidins escape digestion, acting as fiber–antioxidant complex, to be fermented by the enzymes of the gut microbiota, releasing low-molecular-weight phenolic compounds linked to them [58]. Previous scientific works demonstrated that the combined activity of Viscozyme and Pronase was able to mimic the enzymatic hydrolysis of intestinal microbiota, releasing phenolic compounds from the coffee melanoidins [15,59]. The released CGAs could exert a local antioxidant effect in protection against colon cancer, modulating colonic population and providing a wide range of benefits after absorption through the epithelial cells. Moreover, as suggested by Bertolino et al. [31], the increased TPC value highlighted in the SCGc samples after the colonic stage could be caused also by the several biotransformations that involve the phenolic compounds during the fermentation process, including deprotonation of the hydroxyl residue present on the aromatic rings of the CGAs.

Although the evaluation of the in vitro bioaccessibility of polyphenols extracted from baked foods enriched with SCG material after simulated GiD has been evaluated in previous works, the activity of the gut microbiota needed to be clarified. Indeed, the protocol used by Martinez-Saez et al. [35] excluded the colonic stage in the analysis, whereas our results highlighted the critical role of this biological site in the release of antioxidant compounds, suggesting their potential advantages for human health

On the other hand, based on the traditional daily consumption of biscuits and considering that SCG is present in low percentages in the formulated cookies, it is unrealistic to achieve concentrations of bioactive compounds capable of generating a health effect only with SCGc consumption. Moreover, another limitation of this work is the novel food status of SCG in the European Union. According to the web-based list, there are no applications pending for used SCG in baked food products [60].

## 5. Conclusions

In summary, this work revealed that SCG material could be used to fortify baked foods with natural bioactive molecules, such as polyphenols, melanoidins, and caffeine. In fact, our study demonstrated that SCG material could be recognized as a great source of polyphenolic compounds, especially CGA. Among the latter, UHPLC-Q-Orbitrap high-resolution mass spectrometry analysis displayed that 5-CQA was the most abundant CGA found in this coffee by-product. Moreover, high content in caffeine was highlighted in this study, which in previous works, has shown pharmacological activity. Data revealed that after the simulated GiD, the bioaccessibility of SCG polyphenols was higher when compared with the control cookies samples. Moreover, the colonic phase could be considered the pivotal biological site in the release of antioxidant compounds from the formulated cookies, suggesting their potential advantages for human health. Therefore, the incorporation of SCG material in baked foods, which we propose in this study could represent an effective opportunity for the coffee industry to valorize coffee waste and to minimize the environmental impact, provides enriched food products with an enhanced biological activity, which may exert potential health benefits. However, further in-depth studies are needed to broaden the understanding and confirm the future application of SCG for food fortification purpose.

## Figures and Tables

**Table 1 foods-10-01837-t001:** UHPLC-MS parameters of the assayed analytes (*n* = 14).

Compound	Chemical	Adduct	RT	Measured	Theoretical	Accuracy
	Formula	Ion	(min)	Mass *(m/z)*	Mass *(m/z)*	(Δ mg/kg)
Quinic acid	C_7_H_12_O_6_	[M−H]^−^	1.12	191.05531	191.05611	−4.19
5-CQA	C_16_H_18_O_9_	[M−H]^−^	3.18	353.08790	353.08780	0.03
4-CQA	C_16_H_18_O_9_	[M−H]^−^	3.19	353.08768	353.08780	−0.34
Caffeic acid	C_9_H_8_O_4_	[M−H]^−^	3.20	179.03442	179.03498	−3.13
Caffeine	C_8_H_10_N_4_O_2_	[M+H]^+^	3.20	195.08757	195.08765	−0.41
3-CQA	C_16_H_18_O_9_	[M−H]^−^	3.22	353.08762	353.08780	−0.51
3-*p*CoQA	C_16_H_18_O_8_	[M−H]^−^	3.31	337.09232	337.09289	−1.69
5-*p*CoQA	C_16_H_18_O_8_	[M−H]^−^	3.32	337.09290	337.09289	0.03
3-FQA	C_17_H_20_O_9_	[M−H]^−^	3.39	367.10309	367.10346	−1.01
4+5-FQA	C_17_H_20_O_9_	[M−H]^−^	3.40	367.10303	367.10346	−1.17
Ferulic acid	C_>10_H_10_O_4_	[M−H]^−^	3.46	193.05017	193.05063	−2.38
*p*-Coumaric acid	C_9_>H_8_O_3_	[M−H]^−^	3.48	163.03934	163.04006	−4.42
3,4-diCQA	C_25_H_24_O_12_	[M−H]^−^	3.50	515.12103	515.11950	2.97
3,5-diCQA	C_25_H_24_O_12_	[M−H]^−^	3.53	515.11993	515.11950	0.83

Abbreviations: CQA, Caffeoylquinic; *p*CoQA: *p*-Coumaroylquinic acid; FQA, Feruloylquinic acid; diCQA, Dicaffeoylquinic acid.

**Table 2 foods-10-01837-t002:** Chlorogenic acids (*n* = 9), phenolic acids (*n* = 4), and caffeine content in spent coffee grounds, spent coffee grounds-enriched cookies, and control cookies samples. Data are displayed as average value (mg/kg) and standard deviation.

Compound	SCG	SCGc	CTc
		Average (mg/kg) ± SD	
3-CQA	405.9 ± 31.9	25.3 ± 2.1	0.2 ± 0.0
4-CQA	521.7 ± 38.3	31.9 ± 3.3	nd
5-CQA	1163.9 ± 58.4	81.6 ± 6.6	0.3 ± 0.0
3-*p*CoQA	3.2 ± 0.1	0.2 ± 0.0	nd
5-*p*CoQA	6.3 ± 0.2	0.3 ± 0.0	nd
3-FQA	29.8 ± 0.9	1.6 ± 0.1	nd
4+5-FQA	176.5 ± 11.5	11.1 ± 0.2	0.1 ± 0.0
3,4-diCQA	22.4 ± 1.6	1.3 ± 0.0	nd
3,5-diCQA	135.9 ± 9.3	8.3 ± 0.1	nd
*p*-Coumaric acid	0.2 ± 0.0	0.1 ± 0.0	nd
Ferulic acid	0.8 ± 0.0	0.1 ± 0.0	nd
Caffeic acid	7.2 ± 0.3	0.5 ± 0.0	nd
Quinic acid	2.1 ± 0.1	0.1 ± 0.0	nd
Caffeine	1193.9 ± 62.3	64.6 ± 7.8	nd
Total CGAs	2465.6 ± 19.5	161.6 ± 2.3	0.6 ± 0.0

Abbreviations: CQA, Caffeoylquinic; *p*CoQA, *p*-Coumaroylquinic acid; FQA, Feruloylquinic acid; diCQA, Dicaffeoylquinic acid; CGA, Chlorogenic acid; SCG, spent coffee ground; SCGc, spent coffee grounds-enriched cookies; CTc, control cookies. Tukey’s test was used to evaluate differences between SCGc and CTc samples considering *p*-value less than 0.05 as significant.

**Table 3 foods-10-01837-t003:** Total phenolic content value in not-digested samples and during the simulate gastrointestinal digestion.

Sample	Digestion Stage	TPC mg GAE/100 g ± SD
SCGc	Not digested	174.4 ± 6.5
	Oral stage	48.3 ± 3.6
	Gastric stage	22.1 ± 4.3
	Duodenal stage	72.5 ± 2.1
	Pronase	76.6 ± 7.3
	Viscozyme L.	91.1 ± 9.4
	Total colonic stage	167.7 ± 8.3
CTc	Not digested	131.6 ± 5.1
	Oral stage	32.4 ± 2.2
	Gastric stage	12.5 ± 1.3
	Duodenal stage	59.2 ± 3.2
	Pronase	57.7 ± 4.9
	Viscozyme L.	59.1 ± 3.5
	Total colonic stage	116.8 ± 4.2
SCG	Not digested	1067.2 ± 57.3

Abbreviations: SCG, spent coffee ground; SCGc, spent coffee grounds-enriched cookies; CTc, control cookies. Tukey’s test was used to evaluate differences between SCGc and CTc samples considering *p*-value less than 0.05 as significant.

**Table 4 foods-10-01837-t004:** Antioxidant activity in not-digested samples and during the simulated gastrointestinal digestion, evaluated by DPPH, FRAP, and ABTS assays.

Sample	Digestion Stage	DPPH	FRAP	ABTS
SCGc	Not digested	13.6 ± 0.4	10.2 ± 0.3	19.4 ± 0.5
	Oral satge	2.6 ± 0.1	2.0 ± 0.1	3.5 ± 0.3
	Gastric stage	2.1 ± 0.1	1.3 ± 0.1	2.3 ± 0.1
	Duodenal stage	5.4 ± 0.2	3.3 ± 0.2	3.6 ± 0.4
	Pronase	5.5 ± 0.3	2.8 ± 0.1	7.4 ± 0.5
	Viscozyme L.	7.7 ± 0.3	4.6 ± 0.3	6.1 ± 0.3
	Total colonic stage	12.4 ± 0.3	7.4 ± 0.2	13.5 ± 0.4
CTc	Not digested	11.2 ± 0.4	8.6 ± 0.3	16.2 ± 0.6
	Oral satge	1.2 ± 0.1	0.9 ± 0.1	2.8 ± 0.3
	Gastric stage	0.6 ± 0.0	0.4 ± 0.0	1.7 ± 0.1
	Duodenal stage	2.1 ± 0.1	0.9 ± 0.1	2.4 ± 0.2
	Pronase	1.8 ± 0.1	1.1 ± 0.2	4.8 ± 0.4
	Viscozyme L.	4.1±0.3	2.6 ± 0.3	3.3 ± 0.3
	Total colonic stage	5.9 ± 0.2	3.7 ± 0.3	8.1 ± 0.3
SCG	Not digested	186.4 ± 12.7	156.7 ± 13.4	203.9 ± 9.5

Abbreviations: SCG, spent coffee ground; SCGc, spent coffee grounds-enriched cookies; CTc, control cookies. Tukey’s test was used to evaluate differences between SCGc and CTc samples considering *p*-value less than 0.05 as significant.

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
