# Peer review of "In Vitro Bioaccessibility and Antioxidant Activity of Polyphenolic Compounds from Spent Coffee Grounds-Enriched Cookies"

_foods, 2021, doi:10.3390/foods10081837_

Round 1

Reviewer 1 Report

The presented manuscript has interesting results. In my opinion, it would be of interest to the food industry and to the readers of Molecules. As the authors have written in their manuscript, the results obtained by them can be useful in the preparation of functional foods.

The discussion of the results generally was well done and was supported by the relevant references. The conclusions support the results obtained and discussed by the authors. Some parts of manuscript needs improvement. My overall impression is that the authors did a great job in the research of biologically active compounds.

There is some comments which are should be considered by Authors.

Comment 1:

  1. a) Dear authors, please be careful when use acronyms.

For example in abstract the CGA is not explained, also please consider to limit acronyms in abstract. Is difficult to follow. AA is not good idea to express as acronyme.

  1. b) I also advise against using acronyms in the section headings (see in Materials and Methods).
  2. c) The sentences should not start with an acronym that is incorrectly in English (Lines 131, 132, 150, 247
  3. d) all acronyms presented in Tables should be explained (if you want to use acronyms, they should be presented in footnotes).
  4. e) please avoid using acronyms in headings of tables (the tables should be self-readable).

Comment 2

There is no statistical analysis presented in the table, please supply tables with that.

Author Response

Manuscript ID: foods-1312124

Title: In Vitro Bioaccessibility and Antioxidant Activity of Polyphenolic Compounds from Spent Coffee Grounds-Enriched Cookies

Reviewer 1

The presented manuscript has interesting results. In my opinion, it would be of interest to the food industry and to the readers of Molecules. As the authors have written in their manuscript, the results obtained by them can be useful in the preparation of functional foods.

The discussion of the results generally was well done and was supported by the relevant references. The conclusions support the results obtained and discussed by the authors. Some parts of manuscript needs improvement. My overall impression is that the authors did a great job in the research of biologically active compounds.

There are some comments which are should be considered by Authors.

 Comment 1:

1) Dear authors, please be careful when use acronyms.

For example in abstract the CGA is not explained, also please consider to limit acronyms in abstract. Is difficult to follow. AA is not good idea to express as acronyme.

- As suggested by Reviewer 1, the authors explained the acronym CGA referring to chlorogenic acid. Moreover, the authors removed the acronym “AA” in the manuscript.

2) I also advise against using acronyms in the section headings (see in Materials and Methods).

 - As suggested by Reviewer 1, the authors removed the acronyms in the section headings.

3) The sentences should not start with an acronym that is incorrectly in English (Lines 131, 132, 150, 247

- As suggested by Reviewer 1, the authors removed acronyms in the lines rightly listed by Reviewer 1.

4) all acronyms presented in Tables should be explained (if you want to use acronyms, they should be presented in footnotes).

- As suggested by Reviewer 1, the authors added the abbreviations below the Tables.

5) please avoid using acronyms in headings of tables (the tables should be self-readable).

- As suggested by Reviewer 1, the authors removed the acronyms in the headings of tables.

6) There is no statistical analysis presented in the table, please supply tables with that.

- As suggested by Reviewer 1, the authors presented the statistical analysis in the tables.

The authors thank Reviewer 1 for evaluating our manuscript.

Reviewer 2 Report

The authors describe an interesting application of a coffee by-product, namely spent coffee grounds, to be used as an ingredient in cookies.

General comment on style: the text is extremely difficult to read due to excessive use of abbreviations

Line 18: it is debatable if the coffee ground is the “most” significant by-product, when more than half of the cherry would be pulp?

Line 26 and throughout: please only report significant decimals considering the method validation data

Line 124: arabica in botanical nomenclature should be lower case, and C. arabica in italics.

Line 128: please give further details on the drying process

Table 2: see comment above regarding rounding. 1163.9 +/- 58.4. Why do you state a decimal when the deviation is as high as 58?

All tables: define abbreviations

Discussion: please provide a more balanced discussion. Also discuss the limitations of the research. Major limitations include the low contents of the substances in the coffee grounds (because the majority has been extracted). Therefor, health relevant concentrations will not be reached, especially considering that the coffee grounds are contained in low percentages in the cookie and cookie consumption per day is not that high. Therefore, a consumer recommendation would probably be: drink the coffee itself if you want to get the bioactive compounds?

Rather than discussing the health relevancy, another interesting line of discussion would be: How do the cookies perform regarding sensory quality? Do they taste better? Do they have better texture/colour etc?

Conclusions: a limitation of the applicability of spent coffee grounds would be its novel food status in the European Union. There are applications for coffee grounds, but they are not yet granted. See Klingel et al: Klingel, T.; Kremer, J.I.; Gottstein, V.; Rajcic de Rezende, T.; Schwarz, S.; Lachenmeier, D.W. A Review of Coffee By-Products Including Leaf, Flower, Cherry, Husk, Silver Skin, and Spent Grounds as Novel Foods within the European Union. Foods 20209, 665. https://doi.org/10.3390/foods9050665

References: check style, e.g. journal abbreviation. Provide DOI.

Line 468: what is “p” before https?

Author Response

Manuscript ID: foods-1312124

Title: In Vitro Bioaccessibility and Antioxidant Activity of Polyphenolic Compounds from Spent Coffee Grounds-Enriched Cookies

Reviewer 2

The authors describe an interesting application of a coffee by-product, namely spent coffee grounds, to be used as an ingredient in cookies.

1) General comment on style: the text is extremely difficult to read due to excessive use of abbreviations

- As suggested by Reviewer 2, the authors removed some acronyms in the manuscript in order to make it more readable.

2) Line 18: it is debatable if the coffee ground is the “most” significant by-product, when more than half of the cherry would be pulp?

- As suggested by Reviewer 2,  the authors changed this sentence as "Spent coffee ground (SCG) is a significant by-product generated by the coffee industry"

3) Line 26 and throughout: please only report significant decimals considering the method validation data

- As rightly suggested by Reviewer 2,  the authors changed the decimal values in lines 26 and 27.

4) Line 124: arabica in botanical nomenclature should be lower case, and C. arabica in italics.

-  As suggested by Reviewer 2, the authors changed the term "C. Arabica L." in "C. arabica L."

5) Line 128: please give further details on the drying process

- As suggested by Reviewer 2,  the authors added the missing information in the manuscript as: Coffee was prepared through an American coffeemaker (Aigostar Chocolate 30HIK, Italy), and then SCG was recovered from the filter, dried in a laboratory oven until the moisture of the material reached a level between 12% and 14.5%.

6) Table 2: see comment above regarding rounding. 1163.9 +/- 58.4. Why do you state a decimal when the deviation is as high as 58?

- As suggested by Reviewer 2, no decimal points were needed. However, to be consistent with other values, the authors decided to report all the results to one decimal place.

7) All tables: define abbreviations

- As suggested by Reviewer 2, the authors added the abbreviations below the Tables.

8) Discussion: please provide a more balanced discussion. Also discuss the limitations of the research. Major limitations include the low contents of the substances in the coffee grounds (because the majority has been extracted). Therefor, health relevant concentrations will not be reached, especially considering that the coffee grounds are contained in low percentages in the cookie and cookie consumption per day is not that high. Therefore, a consumer recommendation would probably be: drink the coffee itself if you want to get the bioactive compounds?

- As suggested by Reviewer 2,  the authors added the missing information in the manuscript as: On the other hand, based on the traditional daily consumption of biscuits and considering that SCG is present in low percentages in the formulated cookies, it is unrealistic to achieve concentrations of bioactive compounds able of generating a health effect only with SCGc consumption.

8) Rather than discussing the health relevancy, another interesting line of discussion would be: How do the cookies perform regarding sensory quality? Do they taste better? Do they have better texture/colour etc?

- As suggested by Reviewer 2,  the authors added the missing information in the manuscript as: Moreover, the authors reported that the taste, texture, colour and overall acceptance of SCG-enriched cookies were comparable to commercial cookies.

10) Conclusions: a limitation of the applicability of spent coffee grounds would be its novel food status in the European Union. There are applications for coffee grounds, but they are not yet granted. See Klingel et al: Klingel, T.; Kremer, J.I.; Gottstein, V.; Rajcic de Rezende, T.; Schwarz, S.; Lachenmeier, D.W. A Review of Coffee By-Products Including Leaf, Flower, Cherry, Husk, Silver Skin, and Spent Grounds as Novel Foods within the European Union. Foods 20209, 665. https://doi.org/10.3390/foods9050665.

- As suggested by Reviewer 2,  the authors added the missing information in the manuscript as: Moreover, another limitation of this work is the novel food status of SCG in the European Union. According to the web-based list, there are no applications pending for used SCG in baked food products.

11) References: check style, e.g. journal abbreviation. Provide DOI.

- As suggested by Reviewer 2,  the authors adjusted the reference part. The DOI is provided by the journal as [CrossRef].

12) Line 468: what is “p” before https?

- As suggested by Reviewer 2, the authors removed the "p" before https from the manuscript.

The authors thank Reviewer 2 for evaluating our manuscript.